# Study design, rationale and methods of the Revitalising Informal Settlements and their Environments (RISE) study: a cluster randomised controlled trial to evaluate environmental and human health impacts of a water-sensitive intervention in informal settlements in Indonesia and Fiji

Karin Leder ![ORCID],[1] John J Openshaw,[2] Pascale Allotey,[3] Ansariadi Ansariadi ![ORCID],[4] S Fiona Barker ![ORCID],[1] Kerrie Burge,[5] Thomas F Clasen ![ORCID],[6] Steven L Chown ![ORCID],[7] Grant A Duffy ![ORCID],[7] Peter A Faber,[7] Genie Fleming,[7] Andrew B Forbes ![ORCID],[1] Matthew French,[8] Chris Greening ![ORCID],[9] Rebekah Henry ![ORCID],[10] Ellen Higginson,[11] David W Johnston ![ORCID],[12] Rachael Lappan,[9] Audrie Lin,[13] Stephen P Luby ![ORCID],[2] David McCarthy ![ORCID],[10] Joanne E O'Toole ![ORCID],[1] Diego Ramirez-Lovering ![ORCID],[14] Daniel D Reidpath ![ORCID],[15] Julie A Simpson ![ORCID],[16] Sheela S Sinharoy ![ORCID],[17] Rohan Sweeney ![ORCID],[12] Ruzka R Taruc,[4] Autiko Tela,[18] Amelia R Turagabeci ![ORCID],[18] Jane Wardani,[8] Tony Wong ![ORCID],[5] Rebekah Brown ![ORCID],[8] on behalf of the RISE Consortium

For numbered affiliations see end of article.

**Correspondence to**
Professor Karin Leder;
karin.leder@monash.edu

## ABSTRACT

**Introduction** Increasing urban populations have led to the growth of informal settlements, with contaminated environments linked to poor human health through a range of interlinked pathways. Here, we describe the design and methods for the Revitalising Informal Settlements and their Environments (RISE) study, a transdisciplinary randomised trial evaluating impacts of an intervention to upgrade urban informal settlements in two Asia-Pacific countries.

**Methods and analysis** RISE is a cluster randomised controlled trial among 12 settlements in Makassar, Indonesia, and 12 in Suva, Fiji. Six settlements in each country have been randomised to receive the intervention at the outset; the remainder will serve as controls and be offered intervention delivery after trial completion. The intervention involves a water-sensitive approach, delivering site-specific, modular, decentralised infrastructure primarily aimed at improving health by decreasing exposure to environmental faecal contamination. Consenting households within each informal settlement site have been enrolled, with longitudinal assessment to involve health and well-being surveys, and human and environmental sampling. Primary outcomes will be evaluated in children under 5 years of age and include prevalence and diversity of gastrointestinal pathogens, abundance and diversity of antimicrobial resistance (AMR) genes in gastrointestinal microorganisms and markers of gastrointestinal inflammation. Diverse secondary outcomes include changes in microbial

### Strengths and limitations of this study

► Revitalising Informal Settlements and their Environments is a transdisciplinary randomised controlled trial (RCT) assessing a range of human and environmental health outcomes in urban informal settlements in Makassar, Indonesia, and Suva, Fiji.
► The intervention involves a participatory co-design process to deliver site-specific, water-sensitive, modular infrastructure to reduce faecal environmental contamination and improve water cycle management.
► Primary health outcomes include objective markers of gastrointestinal health in children under 5 years; secondary outcomes include measures of prevalence and diversity of pathogens and antimicrobial resistance markers in both environmental and human samples, as well as ecological, psychological, social and economic impacts.
► The RCT design enables settlement-level cluster randomisation, but blinding to the intervention is not possible.
► Logistic constraints include the number and size of communities enrolled, which limits external validity of findings.

contamination; abundance and diversity of pathogens and AMR genes in environmental samples; impacts on ecological biodiversity and microclimates; mosquito vector

abundance; anthropometric assessments, nutrition markers and systemic inflammation in children; caregiver-reported and self-reported health symptoms and healthcare utilisation; and measures of individual and community psychological, emotional and economic well-being. The study aims to provide proof-of-concept evidence to inform policies on upgrading of informal settlements to improve environments and human health and well-being.

**Ethics** Study protocols have been approved by ethics boards at Monash University, Fiji National University and Hasanuddin University.

**Trial registration number** ACTRN12618000633280; Pre-results.

## INTRODUCTION

Informal settlements are home to more than a billion people, mostly in rapidly growing urban areas of low-income and middle-income countries. With predictions suggesting that up to three billion people could be living in urban informal settlements by 2050,[1] it is imperative to address vulnerabilities faced by informal settlement communities. Informal settlements are often located in inhospitable parts of cities prone to flooding, residents frequently face insecure land tenure, and communities often have inadequate provision of essential centralised urban services including clean water and appropriate sanitation facilities.[2] The confluence of factors faced by residents of informal settlements living in poor conditions results in exposure to compromised ecological conditions and environmental contamination with pathogens and disease vectors, which leads to deleterious impacts on health and well-being.[3]

The Sustainable Development Goals (SDGs), a set of aspirational goals and targets for attaining sustainable development by 2030, envision that urban communities will become more inclusive, safe, resilient and sustainable.[4] Reducing environmental exposure to infectious agents transmitted in human faeces is specifically addressed in the SDGs (SDG 6.1 and 6.2); given the impact of such exposures on the burden of disease, they are also implied as a priority by the health goals. However, conventional engineering solutions for water and sanitation—reticulated water and sewerage systems—often do not reach informal urban settlements,[2 5 6] mainly as a result of the economic poverty, legal land tenure issues and lack of political power among residents in these communities. Low-cost solutions, such as household water treatment and on-site sanitation, have had mixed results in recent trials.[7–10] Interventions attempting to address the adverse conditions must therefore consider the complex exposure pathways that link physical aspects of the environment to human health outcomes, and accordingly must interrupt multifaceted vulnerabilities.

The Revitalising Informal Settlements and their Environments (RISE) trial will implement an intervention designed to address these challenges. The RISE study (www.rise-program.org)[11] will collaborate with communities to design and implement a decentralised wastewater infrastructure, which integrates sustainable, water-sensitive technology into buildings and

landscapes. The infrastructure will be implemented at dwelling, neighbourhood and precinct scales. The focus of the RISE water-sensitive intervention is to reduce direct contact with faecal contaminants among informal settlement residents. Delivery of conventional water and sanitation services, namely toilets and hand basins, is incorporated into the RISE study, but in contrast to traditional water, sanitation and hygiene (WASH) approaches, the intervention extends beyond promotion of improved sanitation, drinking water supplies and handwashing practices.[7–10] Instead, the RISE study will also include settlement-scale infrastructure delivery to address environmental contamination from inadequate attention to management of faecal waste and poorly maintained septic systems, both of which are likely to contribute to suboptimal health impacts in high-density living settings such as occurs in urban informal settlements. The RISE study will also concurrently address upgrading of physical access within communities to limit residents' exposure to environmental pollutants, improve overall water cycle management, diversify water source supplies and attend to water drainage and flood management. This holistic, settlement-scale approach aligns with recent calls for 'transformative WASH' or 'WASH-plus' solutions, incorporating a more comprehensive whole-of-systems framing to address a major planetary health challenge.[12 13]

Here, we describe the rationale of the RISE intervention and the study design and methods of a cluster randomised controlled trial (RCT) to assess the impact of the intervention on the environment and human health and well-being. The cluster RCT is being conducted in Makassar, Indonesia, and Suva, Fiji, in cooperation with local and national governments.

## HYPOTHESES AND AIMS

The hypothesised sequence of change in the RISE study is that implementing a water-sensitive intervention in informal settlements leads to improved physical environments, which in turn results in reduced human exposure to pathogenic faecal contamination and flooding hazards, resulting in improved human and ecological health and well-being (figure 1). Specifically, we hypothesise that the intervention will reduce environmental contamination with human faeces. We further hypothesise that this will reduce residents' exposure to gastrointestinal pathogens that cause diarrhoeal disease, poor gastrointestinal function and nutritional deficits, which particularly impacts young children living in informal settlement conditions, as well as reducing exposure to faecal sources of antimicrobial resistance (AMR) genes. Our hypotheses align with the planetary health approach,[14] acknowledging the complex interplay between multiple factors which include health, environmental conditions, urbanisation, water and sanitation management, gender and socioeconomic equity, and climate change.

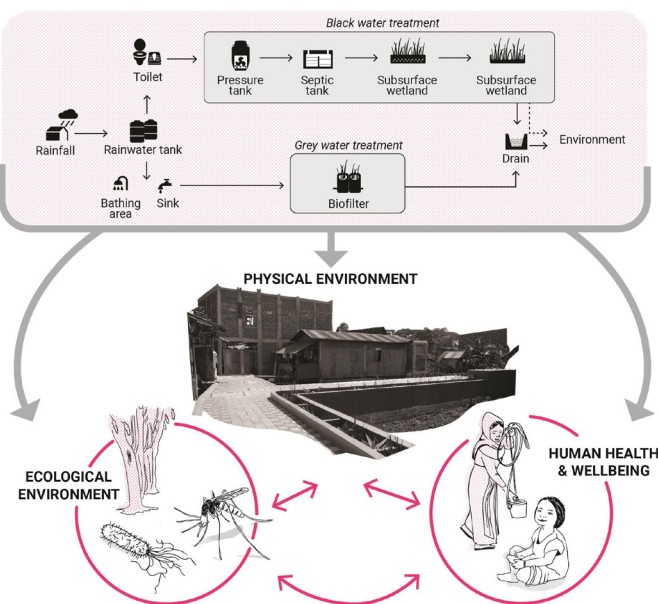

**Figure 1** The Revitalising Informal Settlements and their Environments study hypothesised theory of impacts. The water-sensitive intervention will improve the physical environmental conditions, resulting in both direct and indirect impacts on ecology and human physical health and well-being.

We have three central scientific questions and aims that align with our hypotheses. (In addition to these three scientific questions, the RISE program includes two other research objectives: (1) to develop novel practices for built environment and community co-design, and (2) to develop the financial and policy frameworks for scaling up the intervention.)

### Aim 1: environment

We will first assess whether the RISE intervention leads to improved environmental characteristics. Urbanisation alters physical environmental conditions such as air temperature, soil contamination and water quality and causes transformations in local biodiversity and changes in ecological communities and their functioning.[15–18] Cities tend to have warmer environments than surrounding areas due to the urban heat island effect,[19 20] and in tropical informal settlements, high housing density and inadequate formal solid, greywater and faecal waste processing pathways result in considerable eutrophication and faecal contamination of soil and water.[21] Increased populations of mammal (eg, rats) and arthropod (eg, mosquito) vectors are often present,[22–24] likely due to a combination of thermal changes, habitat modification, resource concentration and high human population densities.[25–27]

Our aim is to assess the impact of delivery of the water-sensitive intervention on the physical, thermal, hydrological and microbiological environments. Changes are expected to make residents less vulnerable to the impacts of flooding, reduce the abundance of vectors, increase the diversity of urban ecological communities and alter the composition of microbial communities. Additionally,

by improving human excreta management strategies, we aim to decrease faecal contamination of the environment, thereby reducing exposure to pathogens and AMR genes in water and soil.

### Aim 2: human health

Our second aim is to assess whether the intervention leads to improved physical health outcomes as a result of improved environmental conditions, as hypothesised above. Prevailing biophysical conditions in informal settlements are conducive to the spread of infectious diseases. Ample vector breeding sites and poor preventative practices lead to mosquito-borne diseases.[28 29] Children suffer from diarrhoea and poor nutrition,[30 31] with contaminated water and soil acting as a source for transmission of pathogens derived from animal, human and environmental origins.[32 33] Antimicrobial-resistant infections are also a growing threat to human health globally,[34] and inadequate faecal waste management leads to environmental contamination, which may promote transmission of multidrug-resistant pathogens between humans and the environment.[35 36] Young children under the age of 5 living in these suboptimal conditions are at especially high risk for undernutrition, diarrhoea and poor growth and development. Further worsening their physical health is the high prevalence of environmental enteric dysfunction, a chronic inflammatory condition of the intestine, which is associated with poor absorption of nutrients, growth faltering and impaired response to live enteric vaccines.[37 38]

By designing and delivering decentralised infrastructure appropriate to and scalable within informal settlements, we aim to improve faecal sludge handling and reduce contact with faecal contamination in order to interrupt the connection between the human faecal stream and the environment. These impacts are expected to decrease children's exposure to gastrointestinal pathogens, as detected by molecular methods in faecal samples, to decrease environmental enteric dysfunction and to decrease the abundance and diversity of AMR markers detected in children's gastrointestinal tract. Flood mitigation and improved access within communities will further decrease human contact with the overall reservoir of environmental pathogens, and falling vector abundance may decrease mosquito-borne disease.

### Aim 3: well-being

Our third aim is to determine whether the intervention affects the physical environment in ways which impact on community well-being and individual subjective psychological and economic well-being. Changes to the physical environment affect inhabitants' lifestyle, mental health, self-image, perceptions of safety and social cohesion.[39–41] They also impact on water security, local economic activities, financial well-being, time allocation for water collection and time available for paid work.[42–46]

We aim to change the physical environment to positively impact inhabitants' evaluative and hedonic subjective

well-being; change how people allocate their time; and improve economic activity by increasing employment opportunities and time spent on income-generating activities. At a community level, we aim to improve residents' perceptions of social cohesion and, motivated through the local co-design approach during intervention development, improve community resilience by enhancing collective efficacy.

### Additional outcomes

In addition to these three key scientific aims, we will evaluate the evidence on the feasibility and impact of the approach to explore the potential for scaling up water-sensitive interventions throughout the Asia-Pacific region through rigorous generation of evidence. We will engage with governments, investors and implementers to determine priorities and economic drivers, and identify barriers to interventions which address the environmental challenges of urban informal settlements.

### STUDY METHODS AND ANALYSIS
### Overview of the study design and timeline

To evaluate the RISE intervention, we are conducting a cluster RCT involving 12 informal settlements in Suva, Fiji, and 12 in Makassar, Indonesia, with clustering at the settlement level. In each country, half of the sites have been randomised as intervention communities and will receive an early upgrade. The remaining communities will serve as controls and will receive only standard, basic hygiene and sanitation messages. Monitoring and assessment will be conducted in all settlements over an approximate 5-year timeline, which will include baseline data collection, the design and build implementation phase of the project, and 2 years of post-build monitoring (figure 2). After completion of the study, the intervention will be implemented in the 12 control settlements.

### Study settings

The study is being conducted across two countries where large portions of the population are subject to inadequate water and sanitation infrastructure. Indonesia and Fiji were purposively selected to represent a diversity of characteristics across the Asia-Pacific, which might influence intervention effects, including climatic factors, water security, tidal inundations, population density, sanitation practices, and socioeconomic and cultural conditions.

Within each country, candidate sites were selected in consultation with local government authorities, intervention funders, research partners, communities and other organisational stakeholders. Formative work consisted of four phases: (1) extensive reconnaissance visits to both cities to understand the range of site conditions and to scope potentially suitable sites; (2) early engagement with local and national governments to determine priorities and preferences, and to ascertain tenure status for each site; (3) household enumeration surveys to determine approximate population size and demographics in each settlement and (4) extensive discussions with community leaders and residents to determine their willingness to participate and priorities.

Makassar, the provincial capital of South Sulawesi, has seen rapid urbanisation and population growth in the last decade, with unplanned city growth resulting in a dramatic rise of informal settlements. The 12 settlements of the RISE project are dispersed across the city and represent a diversity of settlement typologies. While the settlements are located in various flood plains and coastal areas, they are all characterised by dense and precariously built housing, little open space, poor vehicular and pedestrian access, and water stressors. Water bodies and soil are typically contaminated with solid waste derived from animal and human sources, and with domestic black and grey wastewater. Seasonal and other flooding is common, requiring makeshift planking for access. Despite some variation in socioeconomic status within and across settlements, sites typically house the most economically vulnerable populations. Households across the settlements predominantly live on small parcels of privately owned land often with one house, or in some instances two or three houses occupied by extended family groups, with varying degrees of land tenure security.

Similar to the settlement sites in Indonesia, the 12 RISE informal settlements in Suva, Fiji, reflect a range of typologies that exist in Fiji and the broader Pacific context. While the urban morphology varies between settlements, they are typically high-to-medium density clusters with some open space and, in some instances, lush vegetation. Some settlements are prone to significant pluvial and fluvial flooding; housing is of varied quality and construction type but is typically characterised by lightweight structures raised off the ground. Tenure status is characteristic

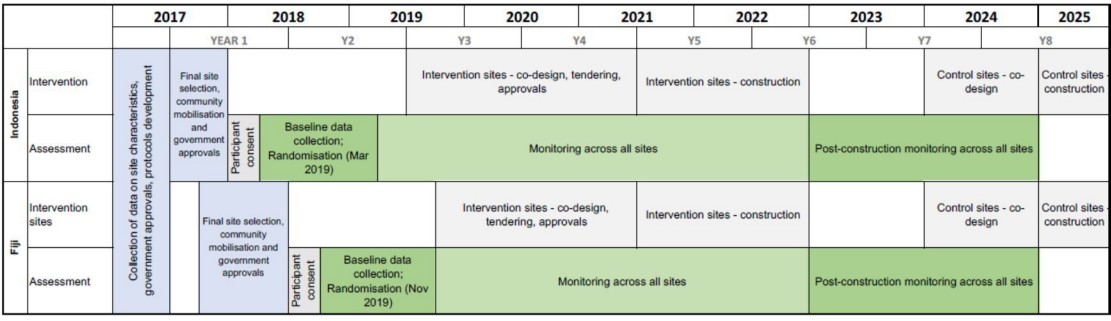

**Figure 2** Anticipated Revitalising Informal Settlements and their Environments study timelines.

| Table 1 | Selection criteria for informal settlements |
|---|---|
| Settlement size, location and demographics | 1. 30–100 houses, ideally all in one consolidated group<br>2. Physically separated from other settlements with clear physical boundaries<br>3. No conditions calling for future permanent relocation (ie, landslide risk)<br>4. Representing the most vulnerable populations of the city<br>5. Location in vulnerable areas with presumed high risk of water-borne disease<br>6. Presence of at least 5–10 children under 5 years |
| Land tenure | 7. Occupants with secure tenure (ownership of the house and land that it sits on, or permission to occupy) |
| Environmental conditions | 8. Presence of water stressors including flooding, poor drainage, limited sanitation and water supply |
| Construction related | 9. Potential for replication and scaling of designs |
| Participation | 10. Settlement leaders and inhabitants consenting for health and environmental assessments and infrastructure modification |

of informal settlements in Fiji and the broader Pacific region with limited private land ownership in the selected communities, instead relying on community leases on state land or informal agreements with traditional land-owning units on native lands.

### Settlement eligibility and selection criteria

Data gathered on potential sites characterised local water supplies, sanitation systems and flooding events and risk. Inspections were conducted at potential sites to confirm feasibility and accessibility. Site locations and sizes were informed by the UN-Habitat informal settlement selection approach.[47] Key inclusion criteria for settlements consisted of location in urban areas of Makassar and Suva, suitability for delivering the planned intervention, and community receptiveness and commitment to study participation. Final selection criteria consisted of settlement size, environmental and construction-related factors (table 1). The key individual inclusion criterion is residence in a settlement enrolled in the trial.

### Site and household recruitment

Community recruitment occurred after extensive discussion with stakeholders, government, community leaders and householders and was performed by trained local community field worker teams. The unit of recruitment for assessment of the intervention is at the household level. We have attempted to recruit and enrol all households within the agreed boundaries of each informal settlement, with written informed consent obtained from heads of households. Householders were able to consent to participation in the overall RISE project with or without consenting to additional individual survey and sampling components.

### Description of the intervention

The water-sensitive approach to upgrading implemented in the RISE study integrates urban design and urban water cycle planning and management, focusing on water conservation and reducing environmental pollution by improving the quality of wastewater and stormwater prior to reuse or discharge to the environment. Implementation of water-sensitive infrastructure will be site-specific, co-designed with each community and adapted to the different biophysical and sociocultural conditions in each informal settlement. The intervention includes specific components collectively designed to reduce residents' exposure to faecal contamination, requiring evaluation of critical faecal contamination and community exposure pathways at each site to determine optimal available technology solutions. Delivery of traditional approaches to water and sanitation services such as private toilets and hand basins are incorporated. Importantly, solutions also include installation of broader decentralised water infrastructure options to control and reduce the release of contaminated sewage and household greywater into the environment, improve flood mitigation, lower exposure to polluted environments within settlements through better access options and provide source water availability through rainwater harvesting and wastewater recycling.

Specific infrastructure options include (figure 3):

► New integrated 'wetpods': these are comprised of a toilet, hand basin and rainwater tank.
► Toilet connections: pressure sewers and traditional collection and treatment systems (pipes, manholes, septic tanks and drains).
► Wastewater treatment: constructed wastewater treatment wetlands for black water,[48–50] and biofilters and biofilter drains for grey water.[51–54]
► Stormwater drainage and treatment: this includes resizing, reprofiling and formalising drains, and swales, rain gardens and constructed stormwater treatment wetlands and permeable paving.[55–58] ;
► Water supply security: rainwater tanks and collection, connection to municipal water supply, water supply disinfection and protection of existing shallow well water supplies.
► Access roads, utility corridors, raised pathways;
► Flood management: including backflow prevention, minor localised spot-filling, minor terrain modification and flood protection walls.

The water-sensitive approach has been shown to successfully address urban water management challenges

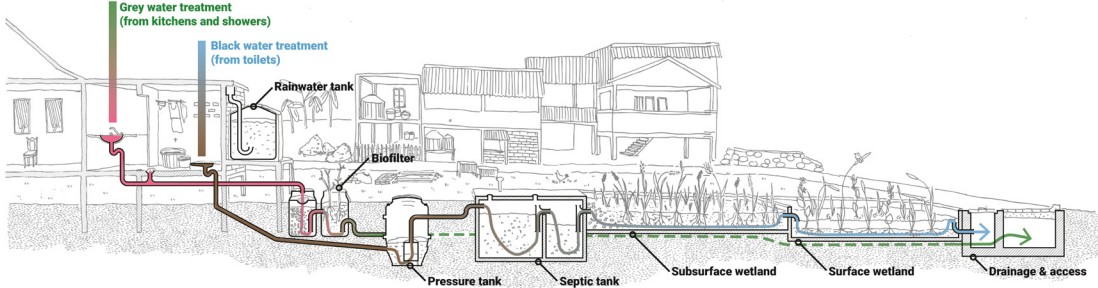

**Figure 3** Schematics of proposed sanitation intervention.

in an integrated way in high-income economies[59–66] and will be adapted for informal settlements in the RISE study. We will initially design and deliver a pilot project in each country to test key water-sensitive technologies and evaluate soils and plants best suited to local conditions. Monitoring system performance from the two pilot communities will underpin adaptive design adjustments to full-scale implementation.

As a sociotechnical intervention, infrastructure design and delivery will be adapted to context-specific social and biophysical characteristics and constraints of each community through a participatory process of co-design. The co-design process will promote engagement and informed decision-making with residents of each settlement, particularly emphasising participation by women, children, elderly, disabled and other groups that may have specific needs. The process is characterised by active community participation to ensure participants' preferences for solutions are heard, including informed choices regarding the physical location of new infrastructure. It brings the international community of researchers and practitioners together with the residents living in the study communities, thereby building understanding and a sense of value of the infrastructure to increase the likelihood of success of long-term operations and maintenance. Communal infrastructure will be delivered by local contractors supervised by the project team. Long-term operation and maintenance responsibilities will be addressed via training and ongoing engagement of community representatives and local authorities. By utilising local construction expertise and engaging and empowering stakeholders at household, community and city levels, we aim to build knowledge and trust with local communities, institutions and governments to ensure the infrastructure is built and maintained for sustainable, long-term benefit.

### Data collection
Data collection is conducted by field workers specifically employed for assessment of trial outcomes; this team is separate from the personnel implementing the intervention. Data collection comprises biospecimen sampling at both the settlement and household levels, field data measurements and household surveys. Specimen collection involves blood and faeces from children under the age of 5 years; environmental soil, water and animal scat samples; and vector trapping, with intended sampling frequency as shown in table 2. Ecological field data assessment involves continuous passive acoustic monitoring, temperature logging and flood assessment including the use of satellite imagery. Anthropometric measures are taken annually among children under the age of 5 years. Surveys are administered quarterly, with different modules asked of varying-aged participants to assess physical health, healthcare utilisation, psychological well-being, financial well-being, time-use and perceptions of social cohesion and collective efficacy. Questions regarding household level data (ie, household member composition, flooding) are administered preferentially to the adult female head of household and questions regarding children under the age of 15 years are preferentially directed to the adult caregiver who has seen them most days in the past 7 days. Questions regarding adult health and well-being, financial well-being, time-use and perceptions of social cohesion and collective efficacy are preferentially administered to: (1) primary caregivers where households contain children under 15 years, and (2) female or (3) male heads for households with no children. If the preferred respondent is not at home, field teams attempt two re-visits. To maintain participation over the trial period and to show appreciation for the time spent by participants on study-related activities, we will provide small culturally appropriate gifts of nominal value, as recommended by our in-country partners.

### Public involvement
Members of enrolled communities are involved in the design and conduct of this trial, including the design and implementation of the intervention as described above.

### OUTCOMES
#### Primary outcomes
We define our first primary outcome as the prevalence and diversity of bacterial, viral, protozoan and helminthic gastrointestinal pathogens in children under 5 years. Additional primary outcomes are (1) the abundance and diversity of AMR genes in gastrointestinal microorganisms of these children; and (2) the concentration of intestinal inflammation and permeability biomarkers in stool representing environmental enteric dysfunction.

**Table 2** Data collection tools and specimens for primary and secondary outcomes

| | Minimum frequency | | | |
|---|---|---|---|---|
| **Primary outcomes** | C | Q | B | A |
| Faeces collection, children <5 years of age for:<br>► Prevalence, diversity and concentration of bacterial, viral, protozoan and helminth gastrointestinal pathogens in children <5 years of age (composite primary outcome)<br>► Concentration of intestinal inflammation markers representing environmental enteropathy in children <5 years of age<br>► Diversity and abundance of AMR genes in gastrointestinal microorganisms of children <5 years of age (composite primary outcome) | | | X | |
| **Secondary outcomes** | | | | |
| *Physical Health Measures* | | | | |
| Blood collection for haemoglobin and blood inflammatory markers, children <5 years | | | | X |
| Anthropometry measures (length-for-age, weight-for-age and weight-for-length z-scores) measure, children <5 years of age | | | | X |
| *Surveys: Household and community demographic, health and well-being* | | | | |
| Household members and demographics | | | | X |
| Household member behaviours and exposures | | | X | |
| Access to sanitation, water and hygiene facilities* | | | | * |
| Caregiver-reported acute symptoms (diarrhoea, respiratory infections and febrile illness), children <5 years of age | | X | | |
| Healthcare system utilisation, children <5 years of age | | X | | |
| Self-reported or caregiver-reported acute symptoms, healthcare system utilisation and subjective general health (5-point rating scale) among adults and children 5–15 years | | | X | |
| Caregiver assessed quality of life (Paediatric Quality of Life Inventory, PedsQL, emotional functioning dimension), children 5–15 years | | | X | |
| Self-reported life satisfaction—multiple domains (10-point rating scale), adults | | | X | |
| Centre for Epidemiologic Studies Depression Scale (CES-D-10), adults | | | | X |
| Self-reported major adverse life events, adults | | | X | |
| Time-use surveys, children 5–15 years and adults* | | | | * |
| Housing tenure+ and collective efficacy (Likert 5-point scales) | | | | X |
| Indicators of household assets and expenditure* | | | | * |
| Intermittent qualitative surveys and focus groups | | NA | | |
| *Measures of environmental faecal contamination* | | | | |
| Soil and water (potable, recreational, wastewater) samples and animal scat samples for chemical and microbiological analyses (including markers of faecal contamination; prevalence, diversity and concentration of pathogens; abundance and diversity of AMR genes) | | | X | |
| *Ecological measures and biodiversity* | | | | |
| Indoor and outdoor thermal data logging | X | | | |
| Passive acoustic monitoring | X | | | |
| Aerial and satellite imagery | | | | * |
| Flood and rain gauges† | | | | X |
| Mosquito trapping | | X | | |
| Rodent trapping† | | | | * |

*First and final years of study.
†Indonesia baseline survey only.
A, annual; AMR, antimicrobial resistance; B, biannual; C, continuous; Q, quarterly.

We will use various molecular approaches to assess the presence and concentration of pathogens, microbial communities and AMR genes in human faecal samples. As indicated below, the same methodologies will be applied for processing of environmental samples, thereby optimising assessment of the impact of the intervention on both environmental and human health outcomes. Total nucleic acids will be extracted from faecal samples in-country. Pathogen levels in samples will be measured by quantitative PCR (qPCR) using TaqMan Array Cards[67 68]

targeting >30 bacterial, viral, protozoan and helminthic pathogens. rRNA gene amplicon sequencing will measure microbial biodiversity. Shotgun metagenomic sequencing will be used to gain information about the functional capabilities of pathogens in samples, including the classification and enumeration of AMR genes. Additionally, over 80 AMR genes will be directly detected by qPCR. Expected environmental enteric dysfunction serological markers include faecal myeloperoxidase,[69] faecal α-1-antitrypsin[70] and faecal neopterin.[71]

## Secondary outcomes
### Human health secondary outcomes
Secondary health outcomes include caregiver-reported and self-reported acute symptoms, healthcare utilisation, inflammatory markers in blood, haemoglobin levels and anthropometry measures. Caregiver-reported and self-reported symptoms will be collected via standardised questionnaires throughout the trial, including the presence of diarrhoea, respiratory and undifferentiated febrile illness over the 7 days prior to the survey visit.[72] Symptoms will be assessed quarterly in children under the age of 5 years and every 6 months in children between 5 and 15 years of age and adults. In all cases, diarrhoea is defined as ≥3 loose or watery stools in a 24-hour period. Healthcare utilisation (visitation to healthcare provider, hospitalisation, antibiotic use) by children under 5 years of age within the 3 months prior to the visit will be surveyed quarterly; for children 5–15 years of age and adults, surveys will be 6 monthly.

Blood from children under 5 years of age will be taken annually by venipuncture. Haemoglobin levels will be measured in the field using a point-of-care device. Plasma and whole blood will be stored for further analyses, which will include markers of nutrition and inflammation. To obtain length-for-age, weight-for-age and weight-for-length z-scores in children under 5 years of age, trained staff will follow standard procedures for anthropometric measurements.[73] Pairs of trained anthropometrists will measure standing (in children between 2 and 5 years of age) or recumbent (less than 2 years of age) length (accurate to 0.1 cm) and weight without clothing (accurate to 0.1 kg) in triplicate. In both cases, the median of the three measurements will be used in the analysis.[74]

### Community and individual well-being secondary outcomes
Secondary outcomes related to individual and community well-being include self-reported general and emotional well-being, time-use, measures of collective efficacy and water security, and indicators of socioeconomic status. We will assess the subjective general health and emotional well-being of children aged 5–15 years using the (parent-proxy) Paediatric Quality of Life Inventory 4.0 Emotional Functioning Scale (PedsQL).[75] Subjective general health, life satisfaction and major life events (ie, family death, serious illness, marriage break-up, bankruptcy, victim of crime) of adults will also be collected 6 monthly.[76 77] Child (age 5–15 years) and adult time-use (eg, minutes/hours spent on different activities in past week) will also be collected. Gender aspects will be examined, including analyses of sex-disaggregated schooling attendance and the intervention's impacts on water collection time and well-being for females in our sample. We will assess collective efficacy to monitor changes in community well-being, trust and feelings of safety.[78] Indicators of socio-economic status (ie, education levels, house building materials, assets owned, bank account, primary activity, time spent working and subjective financial well-being) will also be collected to facilitate distributional analyses of any observed impacts. We will also estimate the costs of the intervention and report on cost-effectiveness and costs-benefits.

### Ecological and environmental secondary outcomes
Secondary outcomes for environmental effects include quarterly chemical and microbiological measures to assess faecal contamination of both soil and water (municipal supply, recreational and wastewater), quarterly measures of vector abundance, and continuous measures of the thermal environment and overall biodiversity. We will compare faecal contamination in water and soil samples across intervention and control settlements. Sample collection and transport will be performed using previously described and optimised methods[79 80] and processed for faecal bacteria quantification.[81] Nucleic acid extraction will be performed in-country via methods that enable simultaneous detection of viruses, bacteria, protozoa and helminths from environmental samples, as confirmed through optimisation experiments already performed. As with human faeces, pathogen presence and concentrations in samples will be measured using TaqMan Array Cards[67 68] and AMR markers will be detected by qPCR and shotgun metagenomic sequencing. rRNA gene amplicon sequencing will be performed to determine microbial community composition and diversity.[82 83] The amplicon outputs will also be used for microbial source tracking using the SourceTracker tool.[84 85] Changes to the thermal environment will be assessed at the settlement level through remote sensed data supported by local thermal data and compared with changes to the built environment over time.[86] Continuous temperature tracking will use thermal dataloggers (Thermochron iButtons). Ten randomly selected houses in each settlement will have thermal data loggers installed and a further five thermal data loggers will be deployed across the settlement to measure ambient temperature and humidity. Rainfall data will be derived from multiple (where available) local government-maintained weather stations enhanced with tipping bucket rain gauges installed to maximise coverage across the study area.

We will examine mosquito species variation and abundance over time based on quarterly trapping results. Mosquitoes will be sampled using BG Sentinel II traps[87] with 15 traps deployed per settlement, inside or outside randomly selected houses. Trap location will remain consistent throughout the assessment period. Mosquitoes

will be identified to species under dissection microscopes using morphological keys.[88–90] Relative abundance and species composition of rodents in communal areas of settlements will also be surveyed in Makassar at baseline using standard traps and a grid-based trapping design to equate trapping effort across sites.[91] The influence of season and physical environmental features, including temperature, humidity and rainfall, on the relative abundance of vector species will be examined.[92 93] Continuous passive acoustic monitoring at settlements will provide a measure of overall biodiversity.[94] The acoustic environment will be monitored using one or two (depending on settlement size) Song Meter SM4 bioacoustic recorders per site for human-audible sound frequencies. Additionally, in Indonesia we will deploy a single Song Meter SM4BAT recorder for ultrasonic frequencies emitted by echolocating bats at each site.

### Referral guidelines
The study will refer participants for treatment to appropriate local healthcare providers in cases of acute malnutrition, anaemia and intestinal helminth infection. WHO weight for length z-scores will be calculated in the field during anthropometry measurements for children under 5 years of age; children meeting WHO/UNICEF criteria for severe malnutrition (weight for length z-score ≤3) will be referred to a local health facility. At times of blood draws, all participating children under the age of 5 years will have their haemoglobin measured in the field. Children with severe anaemia (haemoglobin <7.0 g/dL) will be referred to a local health facility. A subset of faeces samples will be immediately assessed for helminths using the Kato-Katz technique, and individuals with positive results will be referred for treatment.

### Randomisation
Randomisation was performed separately for each city: March 2019 in Makassar and November 2019 in Suva (figure 2). Randomisation occurred following settlement recruitment and completion of 12 months of baseline surveys. Settlements (clusters) have been randomly assigned to either intervention or control groups, with equal numbers in each group. Covariate-constrained randomisation was utilised using the cycrand package in Stata V.15 to achieve balance between the intervention and control groups on key baseline factors determined a priori.

For Makassar, intervention and control groups were first balanced on number of children aged under 5 years (four settlements in each group: ≤10, 11–30 and >30 children) and flood risk (high and low, six settlements in each), producing 90 allocations. Subsequently, the 30 allocations most imbalanced on average asset score were removed, with the score determined from each household's ownership of selected assets. For Suva, intervention and control groups were first balanced on number of children aged under 5 years (<40 children, eight settlements; >40 children, four settlements) and flood risk and

site contamination grouping (two extreme risk, two high risk, and eight medium risk settlements). This resulted in 120 allocations, from which the 30 most imbalanced allocations of asset score were removed, followed by the 30 most imbalanced allocations of average number of children under 5 years. For each of the cities, this resulted in the 60 best balanced allocations being used for the randomisation (total 3600 (60×60) possible allocations for the trial). A randomisation ceremony was held in both cities, where a child from the pilot community selected a ping pong ball from a large glass jar that contained 60 ping pong balls numbered 1–60, representing the 60 determined random allocations that achieved the balance criteria specified above.

### Sample size
The choice of 12 settlements per city, each of ~55 dwellings with five to six people per dwelling, has been made to balance the need for statistical power with the constraints of construction logistics and cost. Statistical power was calculated using formulae for cluster RCTs with repeated assessments of a cohort[95] (and confirmed by numerical simulation using generalised linear mixed models) for the three primary health outcomes assessed in children aged under 5 years: (1) prevalence of gastrointestinal pathogens and diversity (number of bacterial, viral and parasitic gastrointestinal pathogens per sample); (2) diversity and abundance of AMR markers and (3) concentration of intestinal inflammation markers. For measures of diversity and abundance of markers of drug resistance in environmental samples, the power calculations assumed the same intracluster correlations as for the health outcomes (see below).

For the health outcomes involving subsampling of an average 30 children under 5 years of age per settlement, there is 80% power to detect, from 6-month postconstruction and beyond, a 40% relative reduction in the prevalence of gastrointestinal pathogens (assuming a baseline prevalence of 25%) and a 31% relative reduction in the average count of enteric pathogens per child (assuming a baseline average count of 1.0 and overdispersion factor of 2; with a baseline of 2.0 the relative reduction is 23%), and at 12 months an absolute difference of 0.30 SDs (and 0.34 SDs for environmental assessments) in the average concentration of intestinal inflammation markers and average number and abundance of AMR markers. These calculations assume intracluster (ie, within-settlement) correlations between: two children within the same time period 0.10[7]; two children in different time periods 0.067; and repeated measurements of the same child 0.10. The relative reduction of 30% has been informed by the findings reported in systematic reviews, which have found reductions of 30%–50% due to water quality interventions.[96 97]

### Statistical analyses
Analyses will be by intention to treat. Descriptive statistics will report characteristics of settlements and of

individuals by randomised arm. Estimation of the effects of the intervention for each outcome will use generalised estimating equations with an exchangeable correlation structure and robust standard errors clustered at settlement level, scaled with a small-sample degrees-of-freedom adjustment.[98] Data from 6-month, 12-month, 18-month and 24-month follow-up visits will be included in the analysis and the intervention effect estimated for the 6-month follow-up timepoint for prevalence and mean number of gastrointestinal pathogens, and at 12-month follow-up for diversity and abundance of AMR markers and for intestinal inflammation markers. For binary and count outcomes, we will use a logarithmic link to estimate ratios of prevalence and mean counts, respectively; for continuous outcomes, the identity link will be used to estimate differences in means. All models will adjust for the baseline covariates, number of children under 5 years, flood risk category (see Randomisation section) and asset score. To enable temporal adjustment, indicator variables for seasonal patterns (eg, rainy vs dry), local rainfall data for each site and an indicator term for any major flood event at a particular settlement in a particular quarter will be included in the models. For households that drop out of the study, we will employ multiple imputation using imputation models that preserve the hierarchical data structure. Additional sensitivity analyses will exclude households who moved into the settlement postconstruction. There will be no adjustment to the p values for the assessment of multiple primary health outcomes; however, all findings will be fully reported and interpreted based on incremental evidence.

We will explore relationships between environmental changes and health outcomes by modelling health outcomes at 12-month, 18-month and 24-month postconstruction, using changes in environmental measures at 6-month, 12-month or 18-month postconstruction, respectively, as principal covariates. We will estimate these effects using generalised estimating equations with preconstruction health and environmental parameters as baseline covariates aggregated at settlement-level and will include interactions of environmental changes with the intervention for effect-modification assessment, together with adjustment for potential confounders.

Laboratory technicians conducting sample analyses will be blinded to the intervention assignment throughout the study duration through the use of barcodes for sample identification. Owing to the nature of the intervention, community participants will not be blinded.

### Study oversight
The RISE study has an independent International Scientific Advisory Panel (ISAP) and a Governance Committee, both of which provide oversight and advice including in relation to any unintended effects of the trial intervention or trial conduct. The ISAP committee will advise on ethical issues that arise. The study also has a Data Advisory Panel, comprised of representatives across Monash University who provide strategic advice and operational

guidance to assure a holistically advised research data strategy. Given the nature of this trial, there are no stopping rules. Results of the intervention implementation in the first 12 settlements will be used to inform and optimise rollout in the control settlements following completion of the trial.

### Limitations, justifications and additional analyses
Water and sanitation trials aiming for community-level intervention coverage pose great challenges for trial designers.[99] The relative lack of RCTs of such interventions reflects multiple inherent difficulties, including randomly allocating networked infrastructure, the long follow-up and large sample sizes required to study health outcomes and the need for a complex engineering and construction phase.[13] The RISE study will deliver site-specific and bespoke interventions to each settlement, and while this is deliberate and a strength of our approach as it will address site-specific needs, it also results in a lack of intervention uniformity across communities. The purposive selection of settlements included and the relatively small size of each community compared with many informal settlement settings, while necessary from a logistics perspective, also limits broad external validity. Furthermore, while we intend to monitor the sustainability of the intervention, the RCT is funded to perform formal monitoring for an initial 2 years postintervention delivery.

Complete settlement revitalisation should ideally encompass a broad suite of measures incorporating comprehensive upgrading of solid waste management; improvements in housing conditions, roads, street lighting and other amenities in communal areas; and delivering a range of other services. However, to test our specific hypothesis (and given logistic and financial constraints), our planned site-specific intervention is restricted in scope to addressing many of these issues predominantly as they relate to our primary focus of delivering improved water and sanitation infrastructure and reduced flooding impacts, in order to minimise human exposure to environmental faecal contamination. For example, attention to solid waste management is important to ensure that the wetlands, which treat the wastewater on-site, will function as expected and not become blocked with solid waste, as this would reduce the ability of wetland plants to remove contaminants from the wastewater and potentially become a breeding site for vectors (ie, mosquitoes).

Studies such as RISE that involve complex infrastructure upgrades face methodological evaluation challenges and cannot be performed in a blinded fashion, thereby potentially introducing biases in survey-based outcomes. For trials aimed at improving gastrointestinal health among children, the reliability of caregiver-reported diarrhoea is highly variable and in unblinded trials is particularly subject to bias.[100] An alternative approach of testing for organisms in human faeces therefore has advantages, but does not reliably

distinguish between non-pathogenic carriage versus causal links of enteropathogens to diarrhoea[101] or impaired nutrient absorption and growth.[102 103] Hence, we will monitor a combination of primary objective outcomes as well as secondary clinical outcome biomarkers to assess children's physical health. However, while we have described our planned methodologies for analytical testing, the multiyear timeline of the RISE trial means that analytical methods may improve during the study (eg, methods for measuring environmental enteric dysfunction), and accordingly we may need to make changes to the specific tests mentioned in this protocol. The results of our initial analyses to identify which pathogens and AMR genes are most prevalent in the study areas will inform decisions regarding optimal approaches for detection throughout the study period.

Defining primary versus secondary aims and outcomes from complex multifactorial community upgrades requires somewhat artificial prioritisation of transdisciplinary effects. This reflects that anticipated impacts of multidimensional interventions on communities occur along an interconnected causal chain rather than in a simple hierarchical and linear cause-and-effect fashion. While we have prespecified our primary outcomes to reflect objective markers of gastrointestinal health in children under the age of 5 years, our fundamentally transdisciplinary intervention and assessment protocol appropriately accounts for the complex interconnections between ecosystem health, environmental contamination, vector abundance, physical health and well-being. Our intervention has potential to improve liveability through improved access and opportunities for use of community spaces, better economic sustainability through improved microeconomies and urban farming opportunities, and climate resilience through the diversification of water supplies and flood management. We will therefore assess a range of additional secondary, although equally important, impacts on environmental and ecological health, as well as on psychological, social and economic well-being outcomes, which we will measure at both the individual and community level. Our multifaceted monitoring approach and our deliberate focus on assessing the links between human and environmental health will enable us to populate a novel conceptual planetary health model with a broad suite of real-world data. This will advance understanding of mechanisms and impacts of interactions between individual health indicators and provide a surveillance and analytical framework for future studies.

There are significant challenges with both implementing and measuring impacts of infrastructure upgrades and for determining which components of an intervention contribute most effectively to observed outcomes.[104–106] Co-design interventions and upgrading programmes have often failed to include a counterfactual or have taken a quasi-experimental approach,[107] and previous studies have often used narrowly focused or proxy markers of effect rather than direct outcome measures.[104–107] Our planned pragmatic cluster RCT approach to rigorously assess intervention effects is a notable strength. However, the RISE programme involves a complex intervention implemented into complex dynamic communities. Thus, we anticipate a range of outcomes, including potentially important consequences that we cannot accurately prespecify. While we can be most confident of the design-based inference from prespecified outcomes, we also foresee contributing broadly to literature on the impact of the interventions across a wider array of outcomes beyond those explicitly mentioned in this protocol, with findings interpreted based on direction, magnitude and incremental evidence. Additionally, data and specimen collection tools not mentioned here will be appended to address supplementary questions, thereby enabling examination of additional health, environmental and social assessments in the participating informal settlements.

## ETHICS AND DISSEMINATION

Ethics review and approval was provided by participating universities and local IRBs, including Monash University Human Research Ethics Committee (Melbourne, Australia; protocol 9396), the Ministry of Research, Technology and Higher Education Ethics Committee of Medical Research at the Faculty of Medicine, Universitas Hasanuddin (Makassar, Indonesia; protocol UH18020110), and the College Human Health Research Ethics Committee (CHREC) at the Fiji Institute of Pacific Health Research (FIPHR) and College of Medicine, Nursing, and Health Sciences at Fiji National University (FNU) (Suva, Fiji; protocol 137.19). Animal ethics for rodent trapping has also been secured (Monash University project ID16351; Universitas Hasanuddin UH18080446). The trial is registered on the Australian New Zealand Clinical Trials Registry (ANZCTR). Prior to providing informed consent, all study settlements, households and caregivers/respondents are given written explanatory statements describing the voluntary nature of study participation and the fact that they can withdraw from any or all of the study components at any time. Verbal consent is affirmed and documented at each subsequent approach prior to proceeding with survey and sample collection. The data collected in the study will be publicly distributed along with metadata and critical documents (ie, protocols and questionnaires) following the publication of the primary results from the trials.

**Author affiliations**
[1]School of Public Health and Preventive Medicine, Faculty of Medicine, Nursing and Health Sciences, Monash University, Melbourne, Victoria, Australia
[2]Infectious Diseases and Geographic Medicine Division, Stanford University, Stanford, California, USA

[3]International Institute for Global Health, United Nations University, Kuala Lumpur, Malaysia

[4]Public Health Faculty, Hasanuddin University, Makassar, Sulawesi Selatan, Indonesia

[5]CRC for Water Sensitive Cities, Monash University, Melbourne, Victoria, Australia

[6]Gangarosa Department of Environmental Health, Rollins School of Public Health, Emory University, Atlanta, Georgia, USA

[7]School of Biological Sciences, Monash University, Melbourne, Victoria, Australia

[8]Monash Sustainable Development Institute, Monash University, Melbourne, Victoria, Australia

[9]Department of Microbiology, Faculty of Medicine, Nursing and Health Sciences, Monash University, Melbourne, Victoria, Australia

[10]Civil Engineering, Monash University, Melbourne, Victoria, Australia

[11]Cambridge Institute for Therapeutic Immunology and Infectious Disease, University of Cambridge, Cambridge, UK

[12]Centre for Health Economics, Monash Business School, Monash University, Melbourne, Victoria, Australia

[13]Division of Epidemiology and Biostatistics, School of Public Health, University of California Berkeley, Berkeley, California, USA

[14]Art, Design and Architecture, Monash University, Caulfield, Victoria, Australia

[15]Monash University - Malaysia Campus, Bandar Sunway, Selangor, Malaysia

[16]Centre for Epidemiology and Biostatistics, Melbourne School of Population and Global Health, University of Melbourne, Melbourne, Victoria, Australia

[17]Hubert Department of Global Health, Rollins School of Public Health, Emory University, Atlanta, Georgia, USA

[18]School of Public Health and Primary Care, Fiji National University, College of Medicine, Nursing and Health Sciences, Tamavua Campus, Suva, Rewa, Fiji

**Acknowledgements** We acknowledge the hard work and dedication of the RISE project field teams in Fiji and Indonesia, and all the study participants. We also thank Dasha Spasjevic for providing figure 3.

**Collaborators** The RISE Consortium: The RISE consortium: Fitrianty Awaluddin (Public Health Faculty, Hasanuddin University, Indonesia), Becky Batagol (Faculty of Law, Monash University, Australia), Lamiya Bata (Civil Engineering, Monash University, Australia), Dieter Bulachi (Medicine, Dentistry and Health Sciences, Melbourne University, Australia), Bruce Cahan (School of Engineering Department of Management Science & Engineering, Stanford University, USA), Brett Davis (Monash Sustainable Development Institute, Monash University, Australia), Mohammed El-Sioufi (Art, Design and Architecture, Monash University, Australia), Dusan Jovanovic (Civil Engineering, Monash University, Australia), Michaela F Prescott (Art, Design and Architecture, Monash University, Australia), Emma Ramsay (g School of Biological Sciences, Monash University, Australia), Briony Rogersh (School of Social Sciences, Monash University, Australia), Maghfira Saifuddaolah (Public Health Faculty, Hasanuddin University, Indonesia), Christelle Schang (Civil Engineering, Monash University), Chi-Wen Tseng (Civil Engineering, Monash University), Revoni Vamosi (School of Public Health & Primary Care, Fiji National University, Fiji), Silivia Vilsoni (School of Public Health & Primary Care, Fiji National University, Fiji), Isoa Vakarewa (Live and Learn, Fiji), Andi Zulkifli (Public Health Faculty, Hasanuddin University, Indonesia).

**Contributors** KL, PA, TFC, SLC, AF, MF, DWJ, DR-L, SPL, DM, DR, JS, TW and RB devised the project's main conceptual ideas; KL, JJO, AA, SFB, KB, TFC, SLC, GAD, PAF, GF, MF, CG, RH, EH, DWJ, RL, AL, SPL, DM, JEO'T, DR-L, SS, RS, RRT, AT, ART, JW, TW and RB contributed to development of protocol details and implementation; AF and JS designed the statistical analysis plan; KL and JJO contributed to drafting the main manuscript; all authors critically reviewed and approved the manuscript.

**Funding** This trial has been primarily funded by the Wellcome Trust 'Our Planet, Our Health' grant (205222/Z/16/Z). The intervention is funded by the Asian Development Bank, the New Zealand Ministry of Foreign Affairs and Trade, and Monash University.

**Disclaimer** The funders have no role in the study design, data collection and analysis, decision to publish or preparation of manuscripts.

**Competing interests** None declared.

**Patient and public involvement** Patients and/or the public were involved in the design, or conduct, or reporting, or dissemination plans of this research. Refer to the Methods section for further details.

**Patient consent for publication** Not required.

**Provenance and peer review** Not commissioned; externally peer reviewed.

**ORCID iDs**
Karin Leder http://orcid.org/0000-0003-1368-1039
Ansariadi Ansariadi http://orcid.org/0000-0002-9692-6136
S Fiona Barker http://orcid.org/0000-0002-9203-5766
Thomas F Clasen http://orcid.org/0000-0003-4062-5788
Steven L Chown http://orcid.org/0000-0001-6069-5105
Grant A Duffy http://orcid.org/0000-0002-9031-8164
Andrew B Forbes http://orcid.org/0000-0003-4269-914X
Chris Greening http://orcid.org/0000-0001-7616-0594
Rebekah Henry http://orcid.org/0000-0001-6530-8557
David W Johnston http://orcid.org/0000-0003-3185-2890
Stephen P Luby http://orcid.org/0000-0001-5385-899X
David McCarthy http://orcid.org/0000-0001-8845-6501
Joanne E O'Toole http://orcid.org/0000-0002-3161-9215
Diego Ramirez-Lovering http://orcid.org/0000-0003-0774-4929
Daniel D Reidpath http://orcid.org/0000-0002-8796-0420
Julie A Simpson http://orcid.org/0000-0002-2660-2013
Sheela S Sinharoy http://orcid.org/0000-0003-3077-3824
Rohan Sweeney http://orcid.org/0000-0002-3243-9523
Amelia R Turagabeci http://orcid.org/0000-0002-4923-7397
Tony Wong http://orcid.org/0000-0001-8649-2816
Rebekah Brown http://orcid.org/0000-0002-8689-7562

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
