## [Reviewer comments · BMJ Open]

ARTICLE DETAILS

TITLE (PROVISIONAL)	Study design, rationale and methods of the Revitalising Informal Settlements and their Environments (RISE) study: a cluster randomised controlled trial to evaluate environmental and human health impacts of a water sensitive intervention in informal settlements in Indonesia and Fiji
AUTHORS	Leder, Karin; Openshaw, John; Allotey, P; Ansariadi, Ansariadi; Barker, S Fiona; Burge, Kerrie; Clasen, Thomas; Chown, Steven; Duffy, Grant; Faber, Peter; Fleming, Genie; Forbes, Andrew; French, Matthew; Greening, Chris; Henry, Rebekah; Higginson, Ellen; Johnston, David W.; Lappan, Rachael; Lin, Audrie; Luby, Stephen; McCarthy, David; O'Toole, Joanne; Ramirez-Lovering, Diego; Reidpath, Daniel; Simpson, Julie; Sinharoy, Sheela; Sweeney, Rohan; Taruc, Ruzka R; Tela, Autiko; Turagabeci, Amelia; Wardani, Jane; Wong, Tony; Brown, Rebekah

VERSION 1 – REVIEW

REVIEWER	David Osrin UCL Institute for Global Health, UK
REVIEW RETURNED	29-Aug-2020

GENERAL COMMENTS	Study design, rationale and methods of the Revitalising Informal Settlements and their Environments (RISE) study: a cluster randomised controlled trial to evaluate environmental and human health impacts of a water sensitive intervention in informal settlements in Indonesia and Fiji Thank you for asking for my comments on this protocol. The trial that it describes has been developed with a great deal of thought and with contributions from a range of experts. I look forward to seeing the outcomes. The cluster randomised design compares 12 intervention with 12 control informal settlements, with 1:1 allocation stratified into two country sites. It allows for two years of follow-up for outcomes after the intervention. The protocol is well written, concise, and generally follows the SPIRIT guidelines. Major comments The expertise brought to bear on the trial, and the quality of the sections on sample size, allocation, and statistical analysis indicate that the trial team will deliver work of high quality.
--

Because of the nature of the intervention, this is a pragmatic trial. And because such co-design interventions are not usually evaluated in this way, the presence of a counterfactual and a trial design is welcome. For these reasons, I don't want to be too pedantic about the details of the trial.

1. Nevertheless, my main concern lies in the specification of primary outcomes. It is usual for trials of complex interventions to have many potential outcomes. The fact that the authors make some play of this ("Defining primary versus secondary aims and outcomes from complex multifactorial community upgrades requires somewhat artificial prioritisation of transdisciplinary effects...", "While we have pre-specified our primary outcomes to reflect objective markers of gastrointestinal health in children under the age of 5 years...") suggests that they share my concerns. A challenge that we have encountered in designing pragmatic cluster randomised controlled trials is the need for commitment on the outcome. As I don't need to point out, the issue is that the number of possibilities makes post-hoc selection of outcomes that support the intervention a risk. There will doubtless be many publications showing that the intervention improved particular outcomes, and it would be good if there could be a 'main' publication with a clear outcome. This is the downside of using the contemporary trial paradigm, which is a response to concerns about cherry-picking and p-hacking (but still doesn't quite prevent spin). The discourse changes, but in recent trials we have tried to get it down to one or two primary outcomes. Some reviewers (not me) insist on one.
2. There are three primary outcomes, each of which is a composite assessed from a fecal sample from children under five: pathogens, inflammatory biomarkers, and antimicrobial resistance genes. These will be collected every 6 months. Analysis will use GEE models modelling health outcomes at 12-, 18- and 24-months after the intervention. The protocol needs to be clearer on specifying these outcomes and their timepoints. What will be measured, how will it be presented, how will composite outcomes be developed, and what will be the definitive time point...
 - a. Fecal pathogens
 - i. The protocol mentions organism diversity, but the sample size section implies that the outcome will be presented in terms of a reduction in prevalence and count of pathogens. How will these be presented and how will they be combined into a composite outcome?
 - ii. What will the timepoint be? From first sample to last, or a trend based on fall?

	 b. Fecal inflammatory biomarkers  i. The sample size section implies that the outcome will be presented in terms of a difference in levels. How will the different marker levels be combined into a composite outcome? ii. What will the timepoint be? From first sample to last, or a trend based on fall? c. Antimicrobial resistance genes  i. The sample size section implies that the outcome will be presented in terms of average number and abundance (the difference being?) of markers at 12 months. How will these be combined into a composite outcome? 3. I don't know how much equipoise the authors have on the likelihood that antimicrobial resistance patterns will change as a result of the intervention. An option is to consider making this a secondary outcome, particularly as it doesn't have quite the same relationship with children's gut health as the other two. Minor comments  1. Governance. 21a of the SPIRIT template mentions various committees. The section on oversight describes them.  a. I presume the Scientific Advisory Panel will have the status of a Trial Steering Committee and the Governance Committee will have the status of a Trial Management Group. b. A Data Advisory Panel is mentioned, but it doesn't sound like a Data Monitoring Committee or Data Safety Monitoring Board, which cluster randomised controlled trials would usually have. Will there be an independent committee who look at the accruing data and advise on responses to it and ethical issues that arise? c. I presume there will be no stopping rules, given the nature of the intervention. The protocol should include a statement about this. 1. I was struck by the authorship, which includes 23 people from Monash University, 2 from Stanford, 2 from Emory, 1 each from UNU, Cambridge, and Melbourne, 2 from Hasanuddin University, and 2 from Fiji National University. This is a large number of
--	---

	authors which I imagine reflects the attempts of the researchers to achieve transdisciplinarity. The HIC:LMIC ratio is 7:1. Given the thoroughness of the choice of sites described on p6 in study settings, and the participatory nature of the intervention, one might have expected more authors from Indonesia and Fiji.  2. P5 line 9: faecal 3. P14 line 23 “Laboratory technicians...” the word “blinded” could be moved to after “analyses will be.” 4. P15 line 24: sentence has an extra “at”. 5. The online trial registration and SPIRIT checklist say what control clusters will receive, but this is not in the protocol: standard hygiene messages and “basic sanitation focused educational intervention.” 6. 16c of the SPIRIT template says that the allocation sequence was generated by statisticians. I can see the idea, but strictly speaking the statistical process generated series of combinations of clusters and the paper says that assignment was done by community children using a tombola method. 7. The protocol should include a sentence or two on retention and loss to follow-up and how it will be dealt with. 8. Presumably some households will have more than one child under 5, which will imply a further level of clustering. The authors might like to comment in the statistical analysis section on how this will be dealt with. 9. It may or may not be that the authors will adjust for multiple comparisons (opinions seem to differ on this), but the protocol should include a statement about it, particularly given the multiple primary outcomes.
--	---

REVIEWER	Alice Sverdlík IIED, UK
REVIEW RETURNED	10-Sep-2020

GENERAL COMMENTS	The protocol is very clear and well-presented, and the study is likely to make very valuable contributions. Below I've just made a few small suggestions (such as to clarify the limitations a bit more carefully) and also the authors might include a bit of additional discussion of gender and past research on upgrading & health. p8 There is no other mention of gender/women in the text --for well-being measures, I'd suggest looking specifically at the intervention's impacts on gender & time poverty, girls' schooling, and women's empowerment -see also Parikh, P., Fu, K., Parikh, H., McRobie, A., & George, G. (2015). Infrastructure provision, gender, and poverty in Indian slums. World Development, 66, 468-486.; Ray, I. (2007). Women, water, and development. Annu. Rev. Environ. Resour., 32, 421-449; Caruso, B. A., Sevilimedu, V., Fung, I. C. H., Patkar, A., & Baker, K. K. (2015). Gender disparities in water, sanitation, and global health. The Lancet, 386(9994), 650-651. p10 This is the only mention of inadequate solid waste management - it seems this is not one of the study's intervention areas and poor SWM would likely contribute to flooding, vectors etc. so I think it should be mentioned as a limitation. Nor does the upgrading intervention encompass other potential sectors such as housing, electricity, etc. so it might be worth acknowledging this as well. p11 Relatedly, I think the authors should discuss the intervention's
---

	improved access/transport initiatives more fully -it's a bit buried in the text as written. Would suggest adding a few more references on health and upgrading, and also briefly discuss the methodological challenges of evaluating past interventions (see below) because this can help to distinguish the study from past research and highlight its contributions more effectively. For instance see Butala, N. M., VanRooyen, M. J., & Patel, R. B. (2010). Improved health outcomes in urban slums through infrastructure upgrading. Social science & medicine, 71(5), 935-940. IDB (2013) https://publications.iadb.org/publications/english/document/Evaluation-of-Slum-Upgrading-Programs-Literature-Review-and-Methodological-Approaches.pdf Corburn, J., & Sverdlik, A. (2017). Slum upgrading and health equity. International journal of environmental research and public health, 14(4), 342. Kramer and Field (2004) https://projects.iq.harvard.edu/files/wcfia/files/field_impact.pdf
--	---

REVIEWER	Daniel Hammett University of Sheffield, UK
REVIEW RETURNED	24-Sep-2020

GENERAL COMMENTS	I found this to be a very well detailed and developed proposal - with the disclosure from the outset that the proposed work is outwith my primary areas of expertise. The framing of the project re: the SDGs, and broader engagements with informal settlement upgrading projects is sound, and demonstrates awareness of recent work in the field. The design of the project is robust and there are clear justifications for decisions made and plans in place to conduct the study. The key SPIRIT items are covered (as per instructions to reviewers), although I would have been interested to see more reflection and discussion as to the blinding practices (perhaps it's simply my misunderstanding of the wording re: 'participants will not be blinded' and how the processes for anonymising data samples etc will be conducted). Further detail here is needed I would suggest in terms of ensuring confidentiality and/or anonymity of household(er) details. As part of this, it would be important to know to what extent any of those involved in collecting samples/data are embedded within the communities and whether any concerns relating to power relations and who-knows-what-about-whom are present? Similarly, it is noted that culturally appropriate gifts will be provided to maintain participation - the giving of payments/gifts for research project involvement remains a topic of debate. If there is scope within the submission here, a little more reflection on the rationale here would be useful. It is noted that the control sample will have the interventions delivered upon completion of the project period: playing devil's advocate here - would this be the case even if evidence from the intervention pointed to a worsening of outcome indicators? It may seem a fatuous comment/question - but there is a serious underlying ethical question here re: responsibilities and ethics of interventions. I am assuming that were this the case, the intervention would not be introduced in the control sites?
---

	With apologies if I overlooked this - but what are the plans for handling incomplete data returns, specifically: what are the contingency plans for population change within the settlement (new arrivals, those leaving, etc)? And linked to this, while consent for involvement has been secured at this stage - what are the plans for ensuring consent on an ongoing basis? Given the duration of the project this seems to be a key area for attention as to a) what information has been given to participants about their right to withdraw from the project/remove consent for the storing of data/samples, b) how this will be communicated and regularly checked throughout the project lifecycle (i.e. at each of the sampling points), etc. Additionally, while the dissemination section outlines that data will be made publicly available: what are the plans for engaging with local communities/participants/other stakeholders in the presentation and discussion of findings? It is clear that these stakeholders have been closely involved in setting up the project, but it is far less clear as to how they will be informed (and able to feedback on or discuss) about the outcomes of the project in a meaningful way. On a very minor note - there is a missing word or grammatical error in the second paragraph of the 'Statistical analysis plan' section.
--	--

VERSION 1 – AUTHOR RESPONSE

Reviewer: 1, David Osrin, UCL Institute for Global Health, UK

Thank you for asking for my comments on this protocol. The trial that it describes has been developed with a great deal of thought and with contributions from a range of experts. I look forward to seeing the outcomes. The cluster randomised design compares 12 intervention with 12 control informal settlements, with 1:1 allocation stratified into two country sites. It allows for two years of follow-up for outcomes after the intervention. The protocol is well written, concise, and generally follows the SPIRIT guidelines.

Major comments

The expertise brought to bear on the trial, and the quality of the sections on sample size, allocation, and statistical analysis indicate that the trial team will deliver work of high quality.

Because of the nature of the intervention, this is a pragmatic trial. And because such codesign interventions are not usually evaluated in this way, the presence of a counterfactual and a trial design is welcome. For these reasons, I don't want to be too pedantic about the details of the trial.

3) Nevertheless, my main concern lies in the specification of primary outcomes. It is usual for trials of complex interventions to have many potential outcomes. The fact that the authors make some play of this ("Defining primary versus secondary aims and outcomes from complex multifactorial community upgrades requires somewhat artificial prioritisation of transdisciplinary effects...", "While we have pre-specified our primary outcomes to reflect objective markers of gastrointestinal health in children under the age of 5 years...") suggests that they share my concerns. A challenge that we have encountered in designing pragmatic cluster randomised controlled trials is the need for commitment on the outcome. As I don't need to point out, the issue is that the number of possibilities makes post-hoc selection of outcomes that support the intervention a risk. There will doubtless be many publications showing that the intervention improved particular outcomes, and it would be good if there could be a 'main' publication with a clear outcome. This is the downside of using the contemporary trial paradigm, which is a response to concerns about cherry-picking and p-hacking (but still doesn't quite prevent spin). The discourse changes, but in recent trials we have tried to get it down to one or two

primary outcomes. Some reviewers (not me) insist on one.

We completely agree with the reviewer's comments, and appreciate the insight shown. Even the process of trial registration, where up to 3 primary outcomes must be chosen, is not optimally fit-for-purpose for a complex multidisciplinary and multi-faceted study such as ours. A strength of RISE is that we are deliberately assessing a range of impacts from a planetary health perspective, rather than restricting our assessment to more limited physical health effects. However, as acknowledged by the reviewer, we have needed to commit to primary outcomes a priori, and have chosen these to be measures which have an effect on children's gastrointestinal health since this aligns with important public health outcomes. Therefore, our primary outcomes sit at one end of a causal chain of inter-related impacts, including environmental effects and effects on community wellbeing, both of which we will also measure. We have other manuscripts in preparation that will explicitly discuss this fundamental conflict raised by the reviewer, and we are developing a model that shows how our primary outcomes sit centrally within a range of other inter-related and complex impacts.

In line with both these remarks and comments from Reviewer 2, we have added the following paragraph to the final paragraph of the Discussion of our manuscript:

"There are significant challenges with both implementing and measuring impacts of infrastructure upgrades, and for determining which components of an intervention contribute most effectively to observed outcomes[1-3]. Co-design interventions and upgrading programs have often failed to include a counterfactual or have taken a quasi-experimental approach[4], and previous studies have often used narrowly focused or proxy markers of effect rather than direct outcome measures[1-4]. Our planned pragmatic cluster-RCT approach to rigorously assess intervention effects is a notable strength. However, the RISE program involves a complex intervention implemented into complex dynamic communities. Thus, we anticipate a range of outcomes, including potentially important consequences that we cannot accurately prespecify. While we can be most confident of the design-based inference from prespecified outcomes, we also foresee contributing broadly to literature on the impact of the interventions across a wider array of outcomes beyond those explicitly mentioned in this protocol, with findings interpreted based on direction, magnitude and incremental evidence.

Additionally, data and specimen collection tools not mentioned here will be appended to address supplementary questions, thereby enabling examination of additional health, environmental and social assessments in the participating informal settlements."

We acknowledge that evaluation of the validity of any non-prespecified inferences will require thoughtful consideration of the evidence for a causal cascade and care with interpretation of any identified statistical associations within the context of multiple comparisons.

4) There are three primary outcomes, each of which is a composite assessed from a fecal sample from children under five: pathogens, inflammatory biomarkers, and antimicrobial resistance genes. These will be collected every 6 months. Analysis will use GEE models modelling health outcomes at 12-, 18- and 24-months after the intervention. The protocol needs to be clearer on specifying these outcomes and their timepoints. What will be measured, how will it be presented, how will composite outcomes be developed, and what will be the definitive time point...

a. Faecal pathogens

i. The protocol mentions organism diversity, but the sample size section implies that the outcome will be presented in terms of a reduction in prevalence and count of pathogens. How will these be presented and how will they be combined into a composite outcome?

The sample size calculations have been based on showing a reduction in detection of pathogens in children's stool samples. There are two composite measures proposed: (i) the prevalence of any bacterial, viral, or parasitic gastrointestinal pathogen per stool sample, and (ii) the total number of bacterial, viral and parasitic gastrointestinal pathogens per sample. These composite measures were chosen because i) they are suited to the TAC analytical approach, which simultaneously detects >30 pathogens per sample; they directly align with our overarching hypothesis of reducing exposure to environmental faecal contamination; and iii) they are simple to interpret. The reduction in prevalence

will be presented as risk ratios comparing the intervention and control groups with 95% confidence intervals, and the reduction in mean number of pathogens per sample (diversity) will be presented as the ratio of mean number of pathogens between the two groups with 95% CIs.

We anticipate that the use of the TAC will also enable semiquantitative results of pathogens by providing estimates of the number of gene copies per unit of sample (such as per gram of faeces or soil, per litre of water etc). We are currently exploring optimal methods of analysis of abundance, and will provide details of methods of analysis in a Statistical Analysis Plan which will be prepared prior to the completion of data collection and prior to unblinding to group assignment.

We have clarified this in the text, under Sample size, as follows:

“Statistical power was calculated using formulae for cluster RCTs with repeated assessmentsfor the three primary health outcomes assessed in children aged under 5 years: i) prevalence of gastrointestinal pathogens and diversity (number of bacterial, viral and parasitic gastrointestinal pathogens per sample);.....”

ii. What will the timepoint be? From first sample to last, or a trend based on fall?

Estimation of the intervention effect will be derived from GEE modelling of the health outcome data collected at 6, 12, 18 and 24 months post intervention completion, where the primary timepoint for the intervention effect is 6 months post-construction for prevalence and mean number of gastrointestinal pathogens per sample. Our primary analyses will assume that there is a constant effect of the intervention evident from 6 months onwards, with supplementary analyses examining whether there are incremental effects over the 24 month period by allowing the effects to differ at each time point. The specified 12-, 18- and 24-months timepoints are for the exploratory analyses of the relationship between environmental measures and health outcomes.

We have now clarified this in the statistical analyses section by adding the following:

“Data from 6-, 12-, 18- and 24-months follow-up visits will be included in the analysis and the intervention effect estimated for the 6 month follow-up timepoint for prevalence and mean number of gastrointestinal pathogens, and at 12 months follow-up for diversity and abundance of AMR markers and for intestinal inflammation markers.”

b. Fecal inflammatory biomarkers i. The sample size section implies that the outcome will be presented in terms of a difference in levels. How will the different marker levels be combined into a composite outcome?

We will measure a range of fecal inflammatory biomarkers (likely myeloperoxidase [MPO], neopterin [NEO] and alpha-1 antitrypsin [AAT]), although we will be guided by the best available evidence on optimal biomarkers at the time of trial analysis. In the currently existing literature, biomarkers are either analysed individually (log-transformed) or are combined into a derived score. We are staying abreast of this evolving field, and recognise that studies to determine best correlates of histopathological, clinical and biomarker data are underway (e.g. [5]), which we will incorporate into our final analytical decisions.

Given uncertainty about the optimal inflammatory biomarkers, we are also still exploring optimal methods for developing a composite outcome. Various methods for calculation of a composite EED score have been proposed, with one commonly used method being that described by Kosek [6]: AAT, MPO, and NEO categories are defined as 0 (<= 25th percentile), 1, (25–75th percentile), or 2 (>= 75th percentile). The EED score can range from 0 (lowest quartile in all categories) to 10 (highest quartile in all categories) according to the formula $2x(\text{AAT category}) + 2x(\text{MPO category}) + 1x(\text{NEO category}) = \text{EED score}$. Other possible approaches include principal components analysis (PCA) or partial least squared (PLS) regression as used in recent observational studies [7], but these need careful assessment before being incorporated into a randomised trial. Once the composite is determined, as a numerically-scaled measure, the analysis will proceed using GEE with exchangeable correlation using all post-intervention time points, and with the sample size/power defined in terms of standardised effect size, meaning the difference between intervention and control

will be presented in terms of SD units of the final composite measure. Upon finalising our analytical plan, we will publish it on Open Science Framework prior to unblinding laboratory results and conducting analyses.

ii. What will the timepoint be? From first sample to last, or a trend based on fall?

As stated above, estimation of the intervention effect will be derived from GEE modelling of the health outcome data collected at 6, 12, 18 and 24 months, where the primary timepoint for the intervention effect is 6 months post-construction for prevalence and number of gastrointestinal pathogens per sample, and at 12 months post-construction for the composite measure of faecal inflammatory biomarkers (and also for diversity and abundance of AMR markers).

c. Antimicrobial resistance genes i. The sample size section implies that the outcome will be presented in terms of average number and abundance (the difference being?) of markers at 12 months. How will these be combined into a composite outcome?

We will be examining for >80 AMR genes via qPCR, so will have the ability to assess differences in the average number (diversity) of AMR genes per sample between intervention and control communities. Absolute abundance (concentration) is more difficult to measure, but we will attempt to assess the difference in relative abundance of the most commonly identified AMR genes in samples via qPCR and metagenomics, and will report whether we see a difference in either of these outcome measures.

ii. I don't know how much equipoise the authors have on the likelihood that antimicrobial resistance patterns will change as a result of the intervention. An option is to consider making this a secondary outcome, particularly as it doesn't have quite the same relationship with children's gut health as the other two.

We acknowledge that the analytical aspects of assessing AMR results are potentially complex and some may consider it less important than our other primary aims. However, from the perspective of answering broader planetary health questions, improved understanding of the contribution of human faecal contamination to environmental AMR prevalence, abundance and diversity has the potential for transformative public health impact. We therefore consider it a priority to assess whether reduced environmental contamination with human faeces, as anticipated following the intervention, impacts on environmental AMR composition, and in turn whether changes seen are reflected in human gastrointestinal AMR carriage. Additionally, if we decrease illness in the intervention communities, we may reduce antibiotic consumption, thereby further reducing environmental contamination with AMR genes derived from human faeces. We therefore consider it important to retain this question as one of our primary outcomes.

Minor comments

5) Governance. 21a of the SPIRIT template mentions various committees. The section on oversight describes them.

I presume the Scientific Advisory Panel will have the status of a Trial Steering Committee and the Governance Committee will have the status of a Trial Management Group.

A Data Advisory Panel is mentioned, but it doesn't sound like a Data Monitoring Committee or Data Safety Monitoring Board, which cluster randomised controlled trials would usually have. Will there be an independent committee who look at the accruing data and advise on responses to it and ethical issues that arise?

Given the nature of this trial, the main issues that need to be continuously monitored are ethical concerns and major unintended adverse effects (e.g. an increase rather than the anticipated reduction in mosquitoes). The independent International Scientific Advisory Panel will be alerted immediately and will assist with addressing any such issues if they arise. Given RISE is not a traditional (medical) trial, this is a more appropriate and expedient approach than having another separate (traditional DSMB) committee.

The following have been added to the “Study oversight” section:

“The ISAP committee will advise on ethical issues that arise.”

In addition, the following details have been added to section #5d and reiterated in section #21a of the SPIRIT checklist:

“The RISE study has an independent International Scientific Advisory Panel (ISAP) and a Governance Committee, both of which provide oversight and advice including in relation to any unintended effects of the trial intervention.

Specifically, ISAP will provide strategic support and scientific guidance to ensure the highest levels of scientific rigour are adopted and followed. This will ensure that results can stand up to international scrutiny and inform policy and investments. ISAP also provides strategic independent advice on the development and implementation of the RISE program, and, where relevant, informs the RISE team of state-of-the-art research methodologies, tools, and best practices. ISAP members will also serve as a DSMB, and will specifically advise on responses to ethical issues or harms if any are detected in the intervention groups. After significant discussion, and given the complex nature of the trial, using a Panel of experts already familiar with the project to serve in this role was deemed a more appropriate and expedient approach than having another separate (traditional DSMB) committee.

The Governance Advisory Panel serves as an advisory body to the Executive Team, specifically providing independent strategic risk and program management advice.

The study also has a Data Advisory Panel, comprised of representatives across Monash University who provide strategic advice and operational guidance to assure a holistically-advised research data strategy. Specifically, this panel provides advice on how data are collected, stored and shared. The Panel will strengthen the strategy through promotion of best practices and utilisation of fit-for-purpose research infrastructure that in-turn ensure research data accelerates research activity, whilst also maintaining data protection and privacy in accordance with ethical and regulatory requirements.”

6) I presume there will be no stopping rules, given the nature of the intervention. The protocol should include a statement about this.

The reviewer is correct. The following have been added to the “Study oversight” section and to section #21b of the SPIRIT checklist:

“Given the nature of this trial, there are no stopping rules. Results of the intervention implementation in the first 12 settlements will be used to inform and optimise rollout in the control settlements following completion of the trial.”

7) I was struck by the authorship, which includes 23 people from Monash University, 2 from Stanford, 2 from Emory, 1 each from UNU, Cambridge, and Melbourne, 2 from Hasanuddin University, and 2 from Fiji National University. This is a large number of authors which I imagine reflects the attempts of the researchers to achieve transdisciplinarity. The HIC:LMIC ratio is 7:1. Given the thoroughness of the choice of sites described on p6 in study settings, and the participatory nature of the intervention, one might have expected more authors from Indonesia and Fiji.

There are many individuals involved in the study (total number of participating individuals is now >180 across many countries), including personnel employed in both Indonesia and Fiji. However, many of the in-country team members were employed after decisions on study design, site selection and protocol development had been completed. Therefore, many authors from our in-country teams (e.g. lab staff, field staff) will be involved in subsequent papers involving results of survey work and sample collection (some manuscripts are already in-preparation) but are not core authors here on this protocol paper (albeit some are listed within our RISE consortium). Additionally, one of the Monash University authors (JW) did start with RISE as a member of the Indonesian team, but has subsequently moved to Monash to complete a RISE-related PhD and has been assigned a Monash affiliation.

8) P5 line 9: faecal
Fixed

9) P14 line 23 “Laboratory technicians...” the word “blinded” could be moved to after “analyses will be.”
Fixed

10) P15 line 24: sentence has an extra “at”.
Fixed

11) The online trial registration and SPIRIT checklist say what control clusters will receive, but this is not in the protocol: standard hygiene messages and “basic sanitation focused educational intervention.”

This information has now been added under STUDY METHODS AND ANALYSIS, Overview of the study design and timeline as follows:

“The remaining communities will serve as controls and will receive only standard, basic hygiene and sanitation messages.”

12) 16c of the SPIRIT template says that the allocation sequence was generated by statisticians. I can see the idea, but strictly speaking the statistical process generated series of combinations of clusters and the paper says that assignment was done by community children using a tombola method.

This has been changed in section 16c of the SPIRIT checklist, which now reads “Statisticians generated a series of allocations of settlements to intervention and control groups satisfying the balance criteria, from which one allocation was randomly chosen by a child from the pilot community in each country (using the tombola method, see details in ‘Randomisation section).”

13) The protocol should include a sentence or two on retention and loss to follow-up and how it will be dealt with.

The in-country teams actively cultivate strong working relationships with the residents within our settlements. These relationships are critical to the successful roll-out of the intervention; therefore, it is anticipated that loss to follow-up will primarily be residents who move out of the settlement, rather than refusal to participate. We are actively tracking movement of settlement residents / participants. At each survey round, we enrol and consent new households / participants if they decide they would like to participate – these are either people who have recently moved into the settlement, or people who did not initially agree to participate but have now decided to do so. As such, we collect data of anyone within our settlement boundaries willing to participate at each time point. We also capture movement of residents within the settlement (from one house to another) and when they depart the settlement.

The following has now been added to the statistical analysis section:

“For households that drop out of the study, we will employ multiple imputation using imputation models that preserve the hierarchical data structure. Additional sensitivity analyses will exclude households who moved into the settlement post-construction.”

14) Presumably some households will have more than one child under 5, which will imply a further level of clustering. The authors might like to comment in the statistical analysis section on how this will be dealt with.

The analysis method uses GEE together with robust standard errors clustered at the settlement level, and therefore accommodates intermediate levels of clustering between the individual and the settlement (e.g. individual->household->street->block->settlement).

15) It may or may not be that the authors will adjust for multiple comparisons (opinions seem to differ on this), but the protocol should include a statement about it, particularly given the multiple primary outcomes.

The reviewer is correct that whether or not to adjust for multiple comparisons is a contentious issue in the statistical literature. We prefer to not adjust for multiple comparisons as the procedures for doing this are focussed on a binary accept/reject of p-values, and further the 'family' of comparisons for which Type I error is controlled is never well defined. Instead we will focus on interpreting the findings from the multiple primary health outcomes based on the direction, magnitude, 95% confidence intervals and incremental evidence. The following has now been added to the statistical analysis section:

"There will be no adjustment to the p-values for the assessment of multiple primary health outcomes, however, all findings will be fully reported and interpreted based on incremental evidence."

Reviewer: 2, Alice Sverdlik, IIED, UK

The protocol is very clear and well-presented, and the study is likely to make very valuable contributions. Below I've just made a few small suggestions (such as to clarify the limitations a bit more carefully) and also the authors might include a bit of additional discussion of gender and past research on upgrading & health.

16) p8 There is no other mention of gender/women in the text --for well-being measures, I'd suggest looking specifically at the intervention's impacts on gender & time poverty, girls' schooling, and women's empowerment -see also Parikh, P., Fu, K., Parikh, H., McRobie, A., & George, G. (2015). Infrastructure provision, gender, and poverty in Indian slums. *World Development*, 66, 468-486.; Ray, I. (2007). Women, water, and development. *Annu. Rev. Environ. Resour.*, 32, 421-449; Caruso, B. A., Sevilimedu, V., Fung, I. C. H., Patkar, A., & Baker, K. K. (2015). Gender disparities in water, sanitation, and global health. *The Lancet*, 386(9994), 650-651.

We have added that gender impacts will be examined under Community and individual wellbeing secondary outcomes as follows:

"Child (age 5-15 years) and adult time-use (e.g. minutes/hours spent on different activities in past week) will also be collected, and gender aspects examined."

We have not elaborated further on gender issues, but we have secured additional funds for an ancillary "Water for Women" project. This add-on work is specifically aimed at measuring and evaluating gender-related impacts and has its own protocol, hence is not specifically included in our core RISE protocol described here.

17) p10 This is the only mention of inadequate solid waste management -it seems this is not one of the study's intervention areas and poor SWM would likely contribute to flooding, vectors etc. so I think it should be mentioned as a limitation. Nor does the upgrading intervention encompass other potential sectors such as housing, electricity, etc. so it might be worth acknowledging this as well. p11 Relatedly, I think the authors should discuss the intervention's improved access/transport initiatives more fully -it's a bit buried in the text as written.

We have added the following paragraph to the Limitations section:

"Complete settlement revitalisation should ideally encompass a broad suite of measures incorporating comprehensive upgrading of solid waste management; improvements in housing conditions, roads, street lighting and other amenities in communal areas; and delivering a range of other services. However, to test our specific hypothesis (and given logistic and financial constraints), our planned site-specific intervention is restricted in scope to addressing many of these issues predominantly as they relate to our primary focus of delivering improved water and sanitation infrastructure and reduced

flooding impacts, in order to minimise human exposure to environmental faecal contamination. For example, attention to solid waste management is important to ensure that the wetlands, which treat the wastewater on-site, will function as expected and not become blocked with solid waste, as this would reduce the ability of wetland plants to remove contaminants from the wastewater and potentially become a breeding site for vectors (i.e. mosquitoes).”

Additionally, this current submission focuses on the protocol for assessing the impacts of the planned intervention. The multifaceted elements of the RISE study have made it necessary for us to deliberately take the approach of addressing various details of this complex, ambitious program in additional manuscripts. We have another paper (in preparation) that will focus solely on the details of the intervention, including its planned scope, road/access issues, and the extent to which solid waste will be addressed.

18) Would suggest adding a few more references on health and upgrading, and also briefly discuss the methodological challenges of evaluating past interventions (see below) because this can help to distinguish the study from past research and highlight its contributions more effectively. For instance see Butala, N. M., VanRooyen, M. J., & Patel, R. B. (2010). Improved health outcomes in urban slums through infrastructure upgrading. *Social science & medicine*, 71(5), 935-940.

IDB (2013) <https://publications.iadb.org/publications/english/document/Evaluation-of-Slum-Upgrading-Programs-Literature-Review-and-Methodological-Approaches.pdf>

Corburn, J., & Sverdlik, A. (2017). Slum upgrading and health equity. *International journal of environmental research and public health*, 14(4), 342.

Kramer and Field (2004) https://projects.iq.harvard.edu/files/wcfia/files/field_impact.pdf

We thank the reviewer for these terrific additional references and the suggestion to include some additional text on health and upgrading. As mentioned in the comments to Reviewer 1 (see point 3 above), we have added a further paragraph to the end of the manuscript which includes these recommended references and addresses this issue.

Reviewer: 3, Daniel Hammett, University of Sheffield, UK

I found this to be a very well detailed and developed proposal - with the disclosure from the outset that the proposed work is outwith my primary areas of expertise. The framing of the project re: the SDGs, and broader engagements with informal settlement upgrading projects is sound, and demonstrates awareness of recent work in the field.

19) The design of the project is robust and there are clear justifications for decisions made and plans in place to conduct the study. The key SPIRIT items are covered (as per instructions to reviewers), although I would have been interested to more reflection and discussion as to the blinding practices (perhaps it's simply my misunderstanding of the wording re: 'participants will not be blinded' and how the processes for anonymising data samples etc will be conducted). Further detail here is needed I would suggest in terms of ensuring confidentiality and/or anonymity of household(er) details. As part of this, it would be important to know to what extent any of those involved in collecting samples/data are embedded within the communities and whether any concerns relating to power relations and who-knows-what-about-whom are present?

Blinding of participants is impossible as it is a physical intervention. This has been clarified in the Statistical analyses section of the manuscript as follows:

“Owing to the nature of the intervention, community participants will not be blinded.” It has also been clarified in section 17b of the SPIRIT checklist, which now reads: “No blinding of participants. It is impossible to blind community participants given this is a physical intervention.”

Section #17a of the SPIRIT checklist has also been updated, with the following text added:

“Unique, numeric identifiers have been generated for houses, households and participants.

Samples: Samples collected by field staff are identified with unique barcodes. Barcodes for human samples do not identify the settlement. Therefore, laboratory staff are blinded to the identity of sample collection location. After sample processing, all samples receive a second unique non-identifiable barcode prior to long-term storage at -80C.

Data: Information about participants (name, date of birth, etc.) is needed by surveyors to conduct household surveys. Paper copies of personal information used by the surveyors is stored in a locked cabinet and will be destroyed when no longer required. Prior to analysis of household survey data, personal details including names, phone numbers and dates of birth will be removed, with data instead linked using the unique identifiers.”

Additionally, the following has been added to section #27 of the SPIRIT checklist:

“Surveyors (community field workers) are independently employed as project staff, and none are from our study sites or from neighbouring settlements. They are all required to sign confidentiality agreements and undergo extensive training in objective survey and sample collection. This training is repeated before each household survey round. No identifiable data will be published. “

20) Similarly, it is noted that culturally appropriate gifts will be provided to maintain participation - the giving of payments/gifts for research project involvement remains a topic of debate. If there is scope within the submission here, a little more reflection on the rationale here would be useful.

We have carefully considered the issue of incentives for study participation and discussed this at length with our in-country partners, who have been key informants in this process. With them, we have documented a policy for the timing and criteria for gift giving, and we maintain an inventory of all gifts dispensed. Our rationale in providing incentives is the relatively long study duration and high frequency of RISE study activities. In providing culturally appropriate 'gifts' of nominal value we consider that we are showing respect and appreciation for the time spent by participants, as well as the potential inconvenience of RISE study activities for them. These are not for inducement, but are to compensate for time lost. We have a documented policy for the timing and criteria for gift giving.

The relevant sentence in the Data Collection section of the manuscript has been updated to include additional detail (albeit brief), and now reads as follows:

“To maintain participation over the trial period and to show appreciation for the time spent by participants on study-related activities, we will provide small culturally appropriate gifts of nominal value, as recommended by our in-country partners.”

This has also been added to section #18b of the SPIRIT checklist.

21) It is noted that the control sample will have the interventions delivered upon completion of the project period: playing devil's advocate here - would this be the case even if evidence from the intervention pointed to a worsening of outcome indicators? It may seem a fatuous comment/question - but there is a serious underlying ethical question here re: responsibilities and ethics of interventions. I am assuming that were this the case, the intervention would not be introduced in the control sites?

This is an important consideration which we have discussed at length as a team. Given that all settlements have been promised an intervention, there would be ethical issues of both proceeding with delivery of the intervention in the face of evidence for major detrimental outcome indicators as well as with failing to upgrade the settlements assigned to the control study arm. If adverse impacts were seen, we would consider how we could use the results of the intervention in the first 12 settlements to inform and optimise rollout in the settlements initially assigned to the control arm.

22) With apologies if I overlooked this - but what are the plans for handling incomplete data returns,

specifically: what are the contingency plans for population change within the settlement (new arrivals, those leaving, etc)?

This has been addressed in #13 above.

23) And linked to this, while consent for involvement has been secured at this stage - what are the plans for ensuring consent on an ongoing basis? Given the duration of the project this seems to be a key area for attention as to a) what information has been given to participants about their right to withdraw from the project/remove consent for the storing of data/samples, b) how this will be communicated and regularly checked throughout the project lifecycle (i.e. at each of the sampling points), etc.

Written explanatory statements describing the voluntary nature of study participation – with the choice of whole-of-study or some-of-study participation, and the fact that withdrawal from any or all of the study components is allowed at any time – has been provided to participants at the time of recruitment, at which time written consent for participation in study component(s) was obtained. Thereafter, verbal consent is affirmed at each subsequent approach – whether participation in survey or sample collection – prior to the activity proceeding, and is documented in the electronic surveys.

The following has been added to the Ethics and Dissemination” section of the manuscript: “Prior to providing informed consent, all study settlements, households, and caregivers/respondents are given written explanatory statements describing the voluntary nature of study participation and the fact that they can withdraw from any or all of the study components at any time. Verbal consent is affirmed and documented at each subsequent approach prior to proceeding with survey and sample collection.”

24) Additionally, while the dissemination section outlines that data will be made publicly available: what are the plans for engaging with local communities/participants/other stakeholders in the presentation and discussion of findings? It is clear that these stakeholders have been closely involved in setting up the project, but it is far less clear as to how they will be informed (and able to feedback on or discuss) about the outcomes of the project in a meaningful way.

We are engaging with local communities, participants and other stakeholders throughout the project in a number of ways (added to the SPIRIT checklist, section #31a).

- Participant feedback:

- Community feedback on preliminary findings is provided through tailored mechanisms, including (i) annual meetings of the Community Engagement Councils (e.g. presenting baseline findings at randomisation events), (ii) household information sheets; (iii) quarterly project newsletters explaining upcoming campaigns, changes to the program, and findings to date; and (iv) Community Engagement Committee meetings where our community field workers communicate the status of research and any significant findings since the last meeting.

- Health findings: abnormal Kato Katz, haemoglobin and anthropometric results are provided in written form to caregivers, with support (including a referral letter and transportation costs) to seek care at local health facilities.

- Government stakeholders: Preliminary findings and results are presented at the Country-level Technical Coordination Committees, which meet 6-monthly and involve all local partners, relevant Ministries and local authorities. This incorporates feedback on sampling campaigns, response rates, and outcomes from project activities. Reports required for the Asian Development Bank (ADB), such as the ADB Knowledge Products, have also served as a vehicle for communicating preliminary outcomes from the intervention design process.

- Global - Feedback on emerging findings and research highlights through monthly newsletters, blog posts (on our website), and the usual additional mechanisms such as conference presentations,

journal articles, etc.

25) On a very minor note - there is a missing word or grammatical error in the second paragraph of the 'Statistical analysis plan' section.

Amended

VERSION 2 – REVIEW

REVIEWER	David Osrin UCL Institute for Global Health, UK
REVIEW RETURNED	10-Nov-2020

GENERAL COMMENTS	Thank you for asking me to look at the revised version of the protocol. The authors have considered all the reviewers' comments and I would support publication at this stage.
---

REVIEWER	Alice Sverdlik Researcher, International Institute for Environment and Development (IIED), UK
REVIEW RETURNED	24-Nov-2020

GENERAL COMMENTS	Many thanks for the thorough response; the adjusted text is clear and thoughtful, including on research ethics and specification of outcomes as requested by other reviewers. My queries were largely addressed already, although the authors' short new phrase 'gender aspects examined' might need a bit of elaboration -for instance, will they examine differences in gender time burdens during water collection, reductions in stress amongst women and girls etc.? As a brief additional query, given that 'planetary health' is a keyword but mentioned only briefly in the text, perhaps the authors can add a couple of sentences about how their work will build on or advance this literature?
---

REVIEWER	Daniel Hammett University of Sheffield, UK
REVIEW RETURNED	11-Nov-2020

GENERAL COMMENTS	Many thanks for your detailed responses to the first round of comments. My initial comments and requests for clarification were all minor and have been fully addressed and demonstrate the detailed and considered design of the project. I have no further requests or comments to make and I believe the current submission to be publishable.
---

VERSION 2 – AUTHOR RESPONSE

Reviewer: 2

Many thanks for the thorough response; the adjusted text is clear and thoughtful, including on research ethics and specification of outcomes as requested by other reviewers. My queries were

largely addressed already, although the authors' short new phrase 'gender aspects examined' might need a bit of elaboration -for instance, will they examine differences in gender time burdens during water collection, reductions in stress amongst women and girls etc.?

We have added the following detail to the Secondary outcomes section:

“Gender aspects will be examined, including analyses of sex-disaggregated schooling attendance and the intervention’s impacts on water collection time and wellbeing for females in our sample.”

As a brief additional query, given that 'planetary health' is a keyword but mentioned only briefly in the text, perhaps the authors can add a couple of sentences about how their work will build on or advance this literature?

In the Introduction, we have added the following:

“This holistic, settlement-scale approach aligns with recent calls for “transformative WASH” or “WASH-plus” solutions, incorporating a more comprehensive whole-of-systems framing to address a major planetary health challenge.”

In the final “justifications and additional analyses” section we have added the following:

“Our multi-faceted monitoring approach and our deliberate focus on assessing the links between human and environmental health will enable us to populate a novel conceptual model of planetary health with a broad suite of real-world data. This will advance understanding of mechanisms and impacts of interactions between individual health indicators, and provide a surveillance and analytical framework for future studies.”